# The Role of Regulatory T Cells in Cancer Treatment Resistance

**DOI:** 10.3390/ijms241814114

**Published:** 2023-09-14

**Authors:** Anna Dąbrowska, Magdalena Grubba, Amar Balihodzic, Olga Szot, Bartosz Kamil Sobocki, Adrian Perdyan

**Affiliations:** 1Student Scientific Circle of Oncology and Radiotherapy, Medical University of Gdansk, 80-210 Gdansk, Poland; 2Division of Oncology, Department of Internal Medicine, Comprehensive Cancer Center Graz, Medical University of Graz, 8036 Graz, Austria; 3BioTechMed-Graz, 8010 Graz, Austria; 43P-Medicine Laboratory, Medical University of Gdansk, 80-210 Gdansk, Poland; 5Department of Biology, Stanford University, Stanford, CA 94305, USA

**Keywords:** regulatory T cell, resistance mechanisms, immunotherapy, chemotherapy, radiotherapy

## Abstract

Despite tremendous progress in cancer treatment in recent years, treatment resistance is still a major challenge for a great number of patients. One of the main causes is regulatory T lymphocytes (Tregs), which suppress excessive inflammatory responses via the secretion of immunosuppressive cytokines and upregulate the immune checkpoints. Their abundance causes an immunosuppressive reprogramming of the tumor environment, which is ideal for tumor growth and drug inefficiency. Hence, regiments that can regain tumor immunogenicity are a promising strategy to overcome Tregs-mediated drug resistance. However, to develop effective therapeutic regimens, it is essential to understand the molecular mechanisms of Treg-mediated resistance. In this article, we gathered a comprehensive summary of the current knowledge on molecular mechanisms and the role of Tregs in cancer treatment resistance, including cancer immunotherapy, targeted therapy, chemotherapy, and radiotherapy.

## 1. Introduction

Regulatory T cells (Tregs) are a phenotypically and functionally heterogeneous group of T lymphocytes [1]. Tregs come in many forms, including both CD4+ and CD8+ T cells, and are essentially differentiated mostly by the expression of specific markers. For instance, CD4+ CD25+ Tregs actively contribute to the maintenance of immune homeostasis and immunological self-tolerance [1]. A more detailed classification of CD4+ Tregs distinguishes subsets such as CD4+CD25+FoxP3−, CD4+CD25−FoxP3^low^, CD4+CD25^hi^CD125^low^, CD4+CD25^hi^CD45RO^hi^, and CD4+CD25+CD62L^low^CD44^hi^ [2]. Forkhead box protein P3 (FoxP3) plays a crucial role in the development and suppressive function of Tregs. It is a significant regulator that distinguishes Tregs from activated CD4+CD25− T cells [3]. CD8+ Tregs also represent a heterogeneous group, which includes the following subsets: CD8+FoxP3+, CD8+CD122+, CD8+CD28−, CD8αα+, CD8+Qa-1-restricted, and CD8+CD45RC^low^. Nevertheless, the number of studies investigating the immunomodulatory properties of the CD8+ Tregs subset remains insufficient [4]. Tregs are responsible for restoring immune homeostasis after an excessive acute inflammatory response, thus preventing the development of chronic inflammation [5]. Mechanistically, their suppressive properties extend to antigen-presenting cells (APCs), natural killer (NK) cells, and effector T lymphocytes (Teffs) (Figure 1) [1]. Moreover, Tregs exploit other mechanisms to control the excessive inflammatory response and evade tumors (Figure 2). Such mechanisms include modulation of the metabolism of IL-2 and ATP, calcium ions disruption, the secretion of inhibitory cytokines such as granzyme B, IL-10, IL-35, and TGF-β, the inhibition of Teffs and APC through kynurenine production and immune checkpoint (ICs) protein expression, and the secretion of extracellular vesicles (Evs) [6].

Tregs are found in peripheral blood, various tissues, and inflammatory regions, including heterogeneous tumor microenvironment (TME) with other immunosuppressive and immunostimulant cells (Figure 3A). Like other T lymphocytes, Tregs express numerous co-signaling receptors that play critical roles in regulating the immune response (Figure 3B). Some of these receptors, including CTLA-4, GITR, OX40, PD-1, ICOS, TIGIT, LAG-3, TIM-3, and 4-1BB, are expressed on Tregs. The high expression of certain receptors by Tregs and cancer tissues contributes to tumor immune evasion [7,8].

## 2. Results and Discussion

### 2.1. Involvement of Tregs in Immunotherapy Resistance

ICs are receptors expressed by immune cells that are significant for T cell functionality. Tumor cells exploit certain interactions with these receptors to maintain immune tolerance [9]. Immune checkpoint inhibitors (ICIs) are cancer immunotherapies that block the receptors on the surface of T-lymphocytes and tumor cells that control immune cell activity. Contemporary, there are three main groups of ICIs: PD-1 inhibitors, PD-L1 inhibitors, and CTLA-4 inhibitors [7].

The introduction of ICIs in cancer treatment was one of the greatest breakthroughs in recent years. However, it quickly turned out that only ~20% of cancer patients respond to the treatment [10,11]. Additionally, a great number of patients will become resistant with time. Multiple resistance mechanisms limit the potential of ICIs, one of which could be the presence of Tregs [12,13]. Specifically, the heterogeneity of TME may play a major role in immunotherapy resistance [14]. Recently, it has been shown that apoptotic Tregs eliminated the effectiveness of PD-L1 blockade in CRC-bearing mice [15]. Moreover, the presence of Tregs results in the immunosuppressive microenvironment, which is a potential resistance mechanism to the ICIs (i.e., an anti-PD-L1 monoclonal antibody (mAb)—atezolizumab used in urothelial carcinoma) [16].

#### 2.1.1. Primary Resistance

Primary resistance to the ICIs is defined as a lack of initial response or a low overall response rate (RR). To enhance the efficacy of ICIs, researchers are exploring the benefits of combination with other ICIs or systemic therapies (e.g., chemotherapy and radiotherapy) and identifying predictive biomarkers, which are critical for predicting positive responses to ICIs [17]. Current predictive biomarkers to ICIs responsiveness, such as total tumor mutational load (TML), density and distribution of CD8+ T lymphocytes, PD-L1 expression, and T cell clonality, have several limitations. For instance, some patients with a high level of TML exhibit ineffective responses, while other patients with a low level of TML may exhibit favorable responses to ICIs [18]. A correlation between the accumulation of FoxP3+ Tregs and poor prognoses in patients with IL12A^lo^TGFβ1^lo^ expression (type A CRC; *p* = 0.038) was found. In patients with IL12A^hi^TGFβ1^hi^ (type B CRC) expression, high levels of FoxP3 were associated with a better prognosis, but not significantly (*p* = 0.34) [19]. Particularly, a higher Tregs/Teffs ratio within tumor tissue is associated with worse outcomes [20]. However, it can be used to predict primary RR to immunotherapy. Moreover, mAb targeting CTLA-4 and PD-1 may elevate the intratumoral Teffs/Tregs ratio, thereby positively affecting the prognosis [21].

#### 2.1.2. Acquired Resistance

Acquired resistance to immunotherapy develops in patients who ultimately experience illness recurrence despite the initial clinical response. Several mechanisms have been identified, including the upregulation of alternative ICs, defective antigen presentation, a lack of IFN-γ response, T cell exclusion [9], and tumor-mediated immunosuppression or exclusion [17]. However, the identification of a specific mechanism is often difficult due to several challenges: the lack of consistent terminology to define and classify acquired resistance; difficulty in collecting optimal tumor samples for analyses; and the limited efficacy of identifying immune resistance mechanisms in the tumor, host, and TME [17].

Nevertheless, several mechanisms have been proposed recently. APLNR, a G-protein-coupled receptor, regulates T cell responses by modulating JAK1 and IFN-γ signaling, which plays a role in acquired resistance to immunotherapy. It has also been shown that activating mutations in Ptpn2 affect resistance to ICIs through resistance to IFN-γ [17]. Ultimately, an increasing number of Tregs contribute to the development of acquired resistance to immunotherapy [22]. Table 1 summarizes other mechanisms of treatment resistance.

Due to Tregs’ involvement in the development of acquired resistance, there are several therapeutic approaches to target Tregs to overcome the resistance. For example, they can be targeted by CD25 antibodies. Radiotherapy combined with the blockade of CD25, PD-L1, and TIM-3 has been shown to be more effective than radiotherapy with anti-PD-L1/TIM-3 alone in HNSCC treatment. However, the caveat is that non-Treg cells also express CD25. Another potential treatment target is FoxP3. It is closely related to the efficiency of Tregs as the expression of FoxP3, and IL-10 is reduced after anti-PD-1 treatment. Furthermore, Mdb2 protein’s binding to TSDR has been linked to TET2-mediated demethylation, leading to increased FoxP3 expression. Importantly, TSDR methylation is more specific than CD25 blockade, which may be a more effective approach for Tregs depletion [22].

### 2.2. PD-1 and CTLA-4 Expression Ratio between Teffs and Tregs

Tregs are characterized by a low expression of PD-1 in blood and non-cancerous tissues, but a high expression in tumors. Interestingly, around 10% of gastric cancer patients treated with an anti-PD-1 mAb experienced increased tumor infiltration of Tregs and disease progression. Hence, to increase the effectiveness of the therapy, it seems reasonable to deplete the Tregs before starting treatment with the anti-PD-1 mAb [1].

The effectiveness of anti-PD-1/PD-L1 and anti-CTLA-4 mAbs in cancer immunotherapy relies on their interactions with FcƔ receptors (FcƔRs). Due to the high surface expression of CTLA-4 on Tregs and the presence of effector myeloid-expressing, activating FcƔRs in the TME, Tregs are preferentially depleted over Teffs cells. CTLA-4 and PD-1 are expressed on Tregs and Teffs differently, distinguishing the mechanisms of these ICs blocking mAbs. The expression of CTLA-4 is higher on the regulatory tumor-infiltrating lymphocytes (TILs), whereas PD-1 is overexpressed on the effector TILs. Therefore, anti-CTLA-4 or PD-1 depletion of regulatory T cells leads to higher or lower Teffs/Tregs ratios [21].

PD-1/PD-L1 plays a crucial role in tumor evasion of the immune response. Thus, the PD-1/PD-L1 axis is a good target for mAbs as its blockade can return Teffs function and increase the Teffs/Tregs ratio. Anti-PD-1 antibodies, such as pembrolizumab, downregulate FoxP3, which leads to tTregs and pTregs suppression in melanoma patients (Figure 4). In addition, combined anti-CTLA-4 and anti-PD-1 therapy has shown higher effectiveness compared to individual treatments, significantly increasing Teffs infiltration and the Teffs/Tregs ratio within the tumor. The proliferation and immunosuppressive properties of Tregs may be enhanced by PD-1 deficiency. Therefore, a careful balance must be struck to maximize the benefits of anti-PD-1 treatments while minimizing the unintended consequences [16]. The Tregs/Teffs ratio has the potential to become a predictor of OS and chemotherapeutic response. A higher ratio of Tregs/Teffs in a tumor is associated with a worse prognosis for various cancers, including ovarian cancer, lung cancer, and melanoma [22].

### 2.3. Upregulation of ICs

The primary goal of immunotherapy is to turn on or off the immune checkpoints associated with immune surveillance [29]. Numerous reports have demonstrated that if a single immune checkpoint is blocked, other ICs in the TME may be overexpressed. This phenomenon has been observed in lung cancer between PD-1 and TIM-3 checkpoints [7]. Cancer cells are protected from immune-cell-mediated death by upregulated ICs in TME [30]. In mice models, it has been shown that the expression of ICs can affect the differentiation and other properties of Tregs by increasing the secretion of suppressive cytokines. Specifically, IL-10 and IL-35 produced by Tregs lead to the upregulation of ICs, including PD-1, TIM-3, LAG-3, TIGIT, and the depletion of tumor-infiltrating CD4+ and CD8+ T cells. Additionally, the upregulation of ICs may indirectly inhibit the activation of Teffs by negatively affecting APC function [22].

PD-1 is highly expressed on activated lymphocytes and is upregulated by TILs. T cells with high PD-1 expression decrease the production of pro-inflammatory cytokines (e.g., IFN-γ and IL-2) and increase the secretion of IL-10 via the upregulation of various inhibitory receptors, including CTLA-4 or TIM-3 [25]. Other factors that upregulate PD-1 expression are TGF-β and dexamethasone (Figure 5A) [25,30]. Therefore, the purpose of the PD-1 blockade is an inversion of T cell depletion [21].

Cancer cells acquire immunosuppressive properties by forming suppressive TME, reducing their immunogenicity and attempting to evade immune surveillance. One of the key pathways they exploit is the PD-1 pathway by upregulating PD-L1 expression [21]. It should be noted that the presence of PD-L1 is frequently associated with a poor prognosis. Studies have shown that PD-L1 can weaken the anticancer response by itself. In contrast, PD-L1 overexpression is associated with a better response to the inhibition of the PD-1/PD-L1 axis [29]. PD-L1 expression is regulated by two main mechanisms—innate and adaptive immune response. The first one causes PD-L1 upregulation through the oncogenic NPM/ALK kinase pathway via STAT3. The second, acquired immune response, regulates the expression through pro-inflammatory cytokines (e.g., IFN-γ) [31]. Moreover, cytotoxic T cells play a significant role in PD-L1 expression. They release IFN-γ into the TME, which stimulates signal transducers and activators of signaling pathways, resulting in an increase in PD-L1 expression [14]. Abnormal oncogenic signaling pathways are another factor that promotes cancer progression and increases PD-L1 expression. MYC, HIF-1α, and HIF-2α overexpression are involved in PD-L1 upregulation in melanoma, NSCLC, and HNSCC [32]. Hypoxia, radiotherapy, and chemotherapy are further factors that increase PD-L1 expression. Hsieh et al. showed that in irradiated CRC cells, activated ATR signaling upregulates PD-L1 and CD47. Moreover, irradiated tumor cells use the DNA repair signaling pathway to increase PD-L1 and CD47 expression. This connection has been observed in other solid tumors [33]. Chemotherapy enhances TGF-β expression, also leading to PD-L1 upregulation (Figure 5B) [34].

In many malignancies, the interaction of PD-L1 on cancer cells and PD-1 on TIL promotes tumor immune evasion. Blocking this interaction with mAbs against PD-1 or PD-L1 can reactivate Teffs proliferation and function. This includes cytokine production, such as IFN-γ and IL-2, which decrease the Treg number by increasing the Teffs/Tregs ratio [16]. The PD-1/PD-L1 axis is involved in the induction of Treg expansion through modulating the Notch pathway and asparaginyl endopeptidase (AEP) inactivation. The Notch signaling pathway, which is important for cell–cell communication, regulates the differentiation and function of Tregs, while AEP is a lysosomal cysteine protease responsible for FoxP3 destabilization and antigen processing in dendritic cells (DCs). Therefore, Notch activation and AEP inactivation through PD-1/PD-L1 axis upregulation enhance the immunosuppressive properties of Tregs. Some in vitro studies showed that PD-L1 coated beads have the potential to convert naive CD4+ T cells into Tregs through the downregulation of Akt, mTOR, and ERK2 and the upregulation of PTEN (Figure 6) [35].

FoxP3+ CD4+ Tregs express CTLA-4, and CD4+ and CD8+ Teffs upregulate it. It improves suppressive Tregs activity, while its absence causes unregulated T cell proliferation [7,36]. Therefore, a moderate upregulation of CTLA-4 expression potentially indicates the activation of Tregs [37]. Ipilimumab, an anti-CTLA-4 mAb, can effectively kill CTLA-4-expressing Tregs in TME through antibody-dependent cellular cytotoxicity in melanoma- and CRC-bearing mice [1]. Combining anti-CTLA-4 blockade with radiation-induced Tregs depression may further enhance the removal of suppressor T cells within tumor tissues [37].

### 2.4. Involvement of Tregs in Chemotherapy and Radiotherapy Resistance

Contemporary, approximately 80 cytotoxic drugs are approved for cancer treatment. They can be divided by the mechanism of action into alkylating agents (e.g., cyclophosphamide (CPA); oxaliplatin (FOLFOX-6)), antimetabolites (e.g., fluorouracil (5-FU)), topoisomerase inhibitors (tubulin/microtubule inhibitors (e.g., paclitaxel and docetaxel)), and DNA binders or cleavers (e.g., bleomycin) [29]. Long-lasting reduction in Tregs was observed after docetaxel administration in NSCLC and after folinic acid/5-FU/FOLFOX-6 in gastric cancer [38,39]. A high dose of CPA depletes the Tregs population, which is consistent with its strong lymphopenic capabilities. A study on mice with fibrosarcoma has shown that Tregs surviving CPA treatment had rapid proliferation after chemotherapy and therefore inhibited the development of antitumor immunity after lymphodepletion [40]. In small amounts over a long period, CPA is capable of selectively reducing the proliferation of Tregs, including those in the TME. This suggests that low-dose CPA administration can enhance antitumor immune responses without strong lymphopenic effects [41]. This finding on CPA extended to other drugs would be a useful tool for chemotherapy antitumor augmentation as well as for lymphodepletion strategies to potentiate adoptive T cell therapy. The majority of studies have reported that patients with a higher percentage of Tregs within the CD4+ T cell population before chemotherapy had worse long-term outcomes. However, Treg abundance has been proposed as a potential predictive biomarker in several cancers, including TNBC and ovarian cancer [22]. Tumor PD-L1 expression increases after platinum-based neoadjuvant chemotherapy, predicting poor clinical outcomes in NSCLC [42]. Nonetheless, high PD-L1 expression can improve RR with the administration of anti-PD1 treatment. Meta-analysis of 22 randomized controlled trials with 4289 patients showed that the combination of ICI and chemotherapy significantly improved ORR and PFS relative to ICI alone in NSCLC [43].

Radiotherapy is an integral part of cancer treatment, but its effectiveness can be compromised by upregulated markers like PD-L1, CTLA-4, TIM-3, and STAT3, which can induce Tregs proliferation, leading to treatment resistance [44,45]. A study on gamma irradiation showed that CTLA-4 was upregulated by a low dose of γ-ray (1.8 Gy), whereas a high dose (30 Gy) decreased CTLA-4 expression and therefore abolished the suppressive capacity of Tregs [45]. This corresponds with a more recent study where high doses (10 Gy) of radiation increased the expression of LAG-3 and decreased the expression of CD25 and CTLA-4 (Figure 7) [46].

TIM-3 is another coinhibitory molecule that was found to be upregulated in response to radiotherapy. A unique feature of TIM-3 is the lack of known inhibitory signaling motifs in its cytoplasmic tail in comparison to classic ICs such as PD-1 and CTLA-4. Sixty percent of all CD4+FOXP3+ TILs co-express TIM-3, according to a study on patients with lung cancer. Tregs do not constitutively express TIM-3. The exception is intratumoral Tregs, suggesting their immune regulatory roles within the TME [16]. During the combined radiotherapy and anti-PD-1 treatment (pembrolizumab) for HNSCC, it was found that TIM-3 was upregulated on CD8+ T cells and Tregs. The expression of IL-10 in TIM-3-positive Tregs is higher than that in TIM-3-negative Tregs, and they have a higher capacity to inhibit the release of IFN-γ and TNF-α by Teffs (Figure 8) [47].

About 60% of all cancer patients will be treated with radiotherapy during their disease. Radiotherapy can increase the immunogenicity of TME through immunogenic cell death, the release of reactive oxygen species (ROS), and damage-associated molecular pattern (DAMPs), therefore transforming “cold” non-immunogenic to a “hot” immune-reactive TME [48]. Despite enhancing immunogenicity, radiotherapy has also been reported to promote immunosuppressive TME by increasing the transcription of HIF-1α, which induces Tregs proliferation, activating latent TGF-β in the TME that polarizes tumor-associated macrophages (TAMs) into immunosuppressive phenotype and converting CD4+ T cells into Tregs. TAMs are one of the main mediators of immunosuppression, next to Tregs [49]. They suppress Teff’s function and promote tumor growth, creating an immunosuppressive environment [49,50]. These properties are typical for alternatively activated M2 macrophages. It has been shown that the presence of Tregs and M2 macrophages is correlated with each other and the stage of nasopharyngeal carcinoma [51]. Tregs and M2 macrophages can mutually increase each other by secreting immunosuppressive cytokines [50]. The presence of M2 macrophages was associated with increased tumor invasion and therefore with a poorer prognosis in various cancers, including nasopharyngeal carcinoma and CRC [50,51]. Using the small animal radiation research platform, it was demonstrated that radiotherapy resulted in an increase in tumor-infiltrating Tregs that, among other receptors, exhibited a higher expression of CTLA-4 compared with Tregs in non-irradiated tumors, including melanoma, CRC, and renal cell carcinoma [52]. A study on rodents revealed that Tregs are more resistant to radioactivity, less susceptible to radiation-induced cell death, and have a higher frequency of repopulation than CD4+Foxp3 cells. However, Tregs that have been irradiated exhibit functional impairment and a diminished suppressive capacity [37]. Similar results were obtained in human T cell samples, which indicated that both nTregs and iTregs are more resistant to cell death by radiation than CD4+ Tconvs. However, pTregs showed a more robust decrease in Foxp3 expression than tTregs, suggesting that they are more sensitive to the effects of radiation [46]. Tregs in radioresistance have been studied in a small number of cancers, but the available results are consistent. An increased presence of Tregs in TME or peripheral blood correlates with poor response to radiotherapy in NSCLC [53]. Furthermore, Tregs can play an important role in modulating radiation resistance in HNSCC [54]. These results are also supported by the fact that Tregs depletion enhances radiotherapy outcomes in breast cancer and CRC, which is a reason for combined Tregs-targeted drugs and radiotherapy [55,56].

### 2.5. Molecular Mechanisms of Treg-Mediated Treatment Resistance

#### 2.5.1. Apoptosis

A study on human ovarian cancer ascites showed that oxidative stress plays a significant role by triggering the apoptosis of Tregs via ROS in TME. This is attributed to lower amounts of the transcription factor NRF2 in Tregs, which is responsible for the regulation of the cellular antioxidant system, resulting in increased susceptibility to ROS in TME [15,36].

CD39+CD73+ live Tregs can convert ATP to adenosine. Thus, Teffs can be inhibited via an adenosinergic pathway. Apoptotic Tregs also mediate suppression via the A2A pathway through the expression of CD39 and CD73. Additionally, they release higher levels of ATP through pannexin-1-dependent channels, which effectively intensify the suppression of Teffs [15,35]. Therefore, CD39, CD73, and pannexin-1-dependent channels are potential targets for novel treatment. The administration of two inhibitors partially eliminated the immunosuppressive effect of apoptotic Tregs and enhanced the efficacy of PD-L1 blockade in CRC (Figure 9) [15].

#### 2.5.2. TME Metabolism

Hypoxia upregulates HIF-1α in various cells of TME, which then promotes the expression of CD39/CD73 and adenosine receptors A2AR and A2BR. That leads to a significant extracellular adenosine concentration and intracellular cAMP accumulation, which create a beneficial environment for Tregs recruitment [57]. Despite cellular adaptations to hypoxia, reduced oxygenation in TME has far-reaching consequences, particularly in terms of chemoresistance. Firstly, intravenous drug delivery limits the efficacy of chemotherapy. Secondly, hypoxia influences cellular uptake, also impairing the effectiveness of anticancer drugs through associated acidity and drug efflux pump expression (e.g., Pgp). Moreover, the lack of oxygen in the hypoxic TME hampers the induction of cytotoxicity by chemotherapeutics [58]. The necrosis of tumor cells during chemotherapy results in an increased extracellular potassium concentration. This subsequently suppresses the Akt/mTOR pathway, preventing the transformation of resting CD4+ T cells into Teffs and promoting the development of Tregs in melanoma-bearing mice (Figure 10) [59]. Table 2 provides a broader overview of the mechanisms of the TME metabolism involved in Treg-mediated treatment resistance.

#### 2.5.3. TGF-β-Dependent Upregulation of Tregs

TGF-β is a multifunctional cytokine produced by various cells, such as fibroblasts, macrophages, platelets, Tregs, and tumor cells (Figure 11—left side). Tregs are the significant source of latent TGF-β isoform (i.e., TGF-β1) and can activate it through the expression of cell surface docking receptor GARP and αv integrins [65]. On the other side, TGF-β, along with IL-2, is essential to induce CD4+ T cells to express FoxP3, which is crucial for Tregs development [66]. High levels of TGF-β expressed by tumor cells contribute to the establishment of the local immunosuppressive environment. It is achieved by blocking naive T cell differentiation into a Th1 effector phenotype, promoting their conversion into the Tregs subset, and abolishing antigen-presenting functions of DCs [30]. Both Tregs and TGF-β expression or activation increases in irradiated tissues, but available data are inconsistent about the underlying mechanism [52]. For instance, in melanoma, kidney cancer, and CRC models, stereotactic radiotherapy elevates the levels of intratumoral Tregs in a non-TGF-β-dependent manner, while in a murine prostate cancer model, TGF-β1 mediates Tregs elevation in response to radiation [52,67]. The regrowth of irradiated tumors was significantly correlated with TGF- β1 levels and Treg accumulation when mouse models of prostate cancer were exposed to sub-lethal doses of radiation. Moreover, the inhibition of TGF-1 led to a reduction in Treg accumulation and tumor regrowth after treatment [67]. It should be underlined that radiation-induced TGF-β production depends on dose, time, and tissue [68]. Further investigation is needed to fully understand the precise role of TGF-β in the accumulation of Tregs in various tumor tissues (e.g., lung, prostate, HNSCC) [68,69]. Chemotherapy effects are correlated with epithelial–mesenchymal transition (EMT), the process that contributes to stem cell generation, anticancer drug resistance, genomic instability, and localized immunosuppression [34]. The mechanisms in which cytotoxic agents (e.g., cisplatin) increase TGF-β expression are led through the activation of intracellular transcriptional effectors SMAD (Figure 11) [70]. Table 3 summarizes other mechanisms of the TGF-β-dependent upregulation of Tregs.

## 3. Materials and Methods

This narrative review was conducted according to the SANRA guidelines (https://researchintegrityjournal.biomedcentral.com/articles/10.1186/s41073-019-0064-8; accessed on 5 October 2022). In October 2022, we performed a search using PubMed, Scopus, and Google Scholar. We used the following search queries: “regulatory T cell”, “treatment resistance”, “immunotherapy”, “chemotherapy”, “radiotherapy”. Additionally, more articles were found in the references section of the included articles. Studies in languages other than English were excluded from this narrative review.

## 4. Conclusions

Tregs play an important role in suppressing the antitumor response due to their intrinsic immunosuppressive properties. A high level of tumor-infiltrating Tregs in TME is a negative prognostic factor for various types of cancer. Understanding the role of this subset of T cells and their interactions with other cells and molecules is crucial for the development of effective therapeutic regimens. In this review, we have collected and presented available information on the essential roles of Tregs in cancer treatment resistance. In addition, we have summarized and explained how available therapeutics and their combinations may enhance Tregs targeting that could overcome Tregs-mediated resistance and eventually improve patient outcomes. For example, combining different ICIs (e.g., anti-CTLA-4 and anti-PD-L1) leads to an increased Teffs to Tregs ratio and enhanced immunogenic cell death. Various therapeutic methods that increase the immunogenicity of tumors (e.g., chemo- or radiotherapy) with ICIs can also increase their overall effectiveness. Moreover, we recognized the crucial molecular mechanisms of Treg-mediated resistance that are potential targets for novel therapeutics. Undoubtedly, numerous combination possibilities will exist in the future. However, Tregs-targeted therapies require further extensive research to develop the most optimal, safe, and effective therapeutic strategies to reduce side effects and overcome cancer treatment resistance.

## Figures and Tables

**Figure 1 ijms-24-14114-f001:**
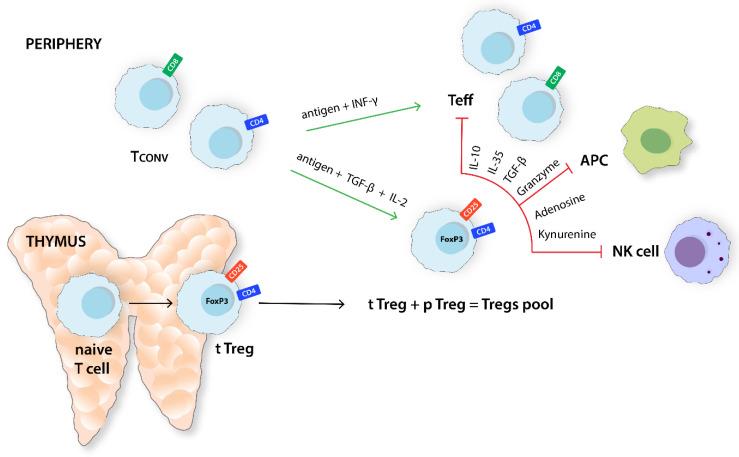
Differentiation and the main function of Tregs. CD4+CD25+FoxP3+Tregs population consists of fractions derived from the thymus and those arising from peripheral Tconv in the presence of environmental antigen stimulation, TGF-β and IL-2. Tregs decrease excessive inflammation, inhibiting the activity of Teffs, APC, and NK cells. Tconv, T conventional cell; IFN-γ, interferon γ; Teff, effector T cell; TGF-β, transforming growth factor β; APC, antigen-presenting cell; NK cell, natural killer cell; FoxP3, forkhead box P3; tTreg, thymus-derived Treg; pTreg, peripherally derived Treg.

**Figure 2 ijms-24-14114-f002:**
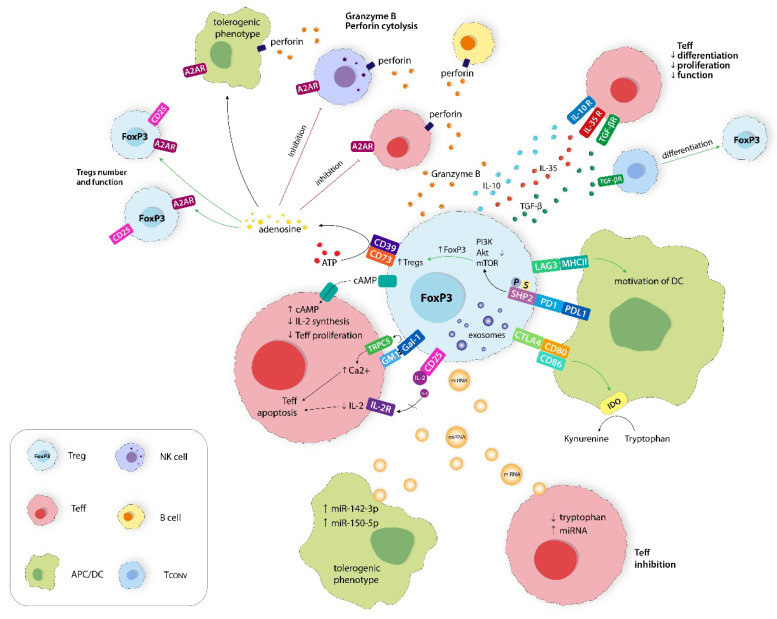
Immunosuppression mechanisms of Tregs are directed to inhibit Teffs, NK cells, B cells, and APC functions. This includes metabolic disruption on Teffs, expression of inhibitory receptors, production of immunosuppressive molecules, and secretion of extracellular vesicles (Evs). A2AR, adenosine 2A receptor; PI3K, phosphoinositide 3-kinase; Akt, cellular homolog of murine thymoma virus Akt8 oncogene; mTOR, mammalian (or mechanistic) target of rapamycin; SHP2, Src homology phosphotyrosyl phosphatase 2; IDO, indolamine 2,3-dioksygenase 1; miRNA, microRNA; Gal-1, Galectin-1; TRPC5, transient receptor potential channel 5; Ca^2+^, calcium ion; GM1, monosialotetrahexosylganglioside; ATP, adenosine triphosphate; cAMP, cyclic adenosine monophosphate; APC/DC, antigen-presenting cell/dendritic cell; Tconv, T conventional lymphocyte.

**Figure 3 ijms-24-14114-f003:**
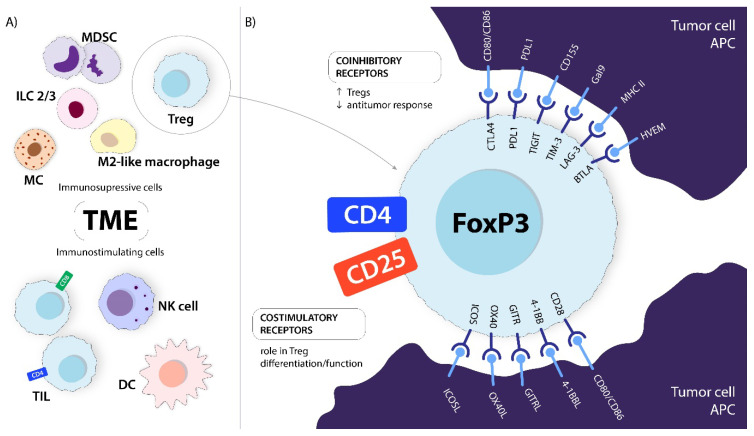
(**A**) Examples of immunosuppressive and immunostimulatory cells in TME. (**B**) Tregs are an immunosuppressive cell subset, which can express multiple co-signaling receptors, both inhibitory and stimulatory, all important for Treg homeostasis. CTLA-4, OX-40, GITR, and BTLA are constitutively expressed by Tregs, while PD-1, TIGIT, TIM-3, LAG-3, and 4-1BB are preferentially overexpressed on tumor-infiltrating Tregs. Activation of coinhibitory receptors through ligation with corresponding ligands leads to enhanced Treg function, resulting in tumor progression. Activation of costimulatory receptors on Tregs can both promote and diminish their suppressive function [9]. MDSC, myeloid-derived suppressor cell; MC, mast cell; ILC2/3, innate lymphoid cell type 2 and 3; M2-like tumor-associated macrophages; TIL, tumor-infiltrating CD4+ and CD8+ lymphocytes; NK, natural killer cell; DC, dendritic cell; BTLA, B, and T lymphocyte attenuator; HVEM, herpesvirus entry mediator; LAG-3, lymphocyte-activation gene 3; MHCII, major histocompatibility complex II; TIM-3, T cell immunoglobulin and mucin-domain containing-3; Gal-9, Galectin 9; TIGIT, T cell immunoreceptor with Ig and ITIM domains; PD-1, programmed cell death; PD-L1, programmed cell death ligand; CTLA-4, cytotoxic T cell antigen 4; 4-1BB/4-1BBL, tumor necrosis factor receptor superfamily 9 and ligand; GITR/GITRL, glucocorticoid-induced TNFR-related protein and ligand; OX40/OX40L, tumor necrosis factor receptor superfamily, member 4/ligand; ICOS/ICOSL, inducible T cell co-stimulator/ligand; APC, antigen-presenting cell.

**Figure 4 ijms-24-14114-f004:**
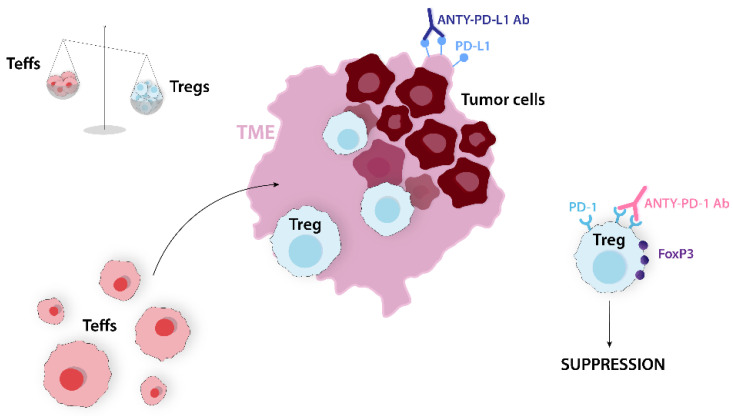
PD-1/PD-L1 axis blockade leads to restoration of Teffs and Tregs suppression, resulting in increased Teffs/Tregs ratio. Anti-PD-1 Ab, anti-PD-1 antibody; anti-PD-L1 Ab, anti-PD-L1 antibody; FoxP3, forkhead box P3; PD-1, programmed cell death; PD-L1, programmed cell death ligand; Teffs, effector T cells; TME, tumor microenvironment; Tregs, regulatory T cells.

**Figure 5 ijms-24-14114-f005:**
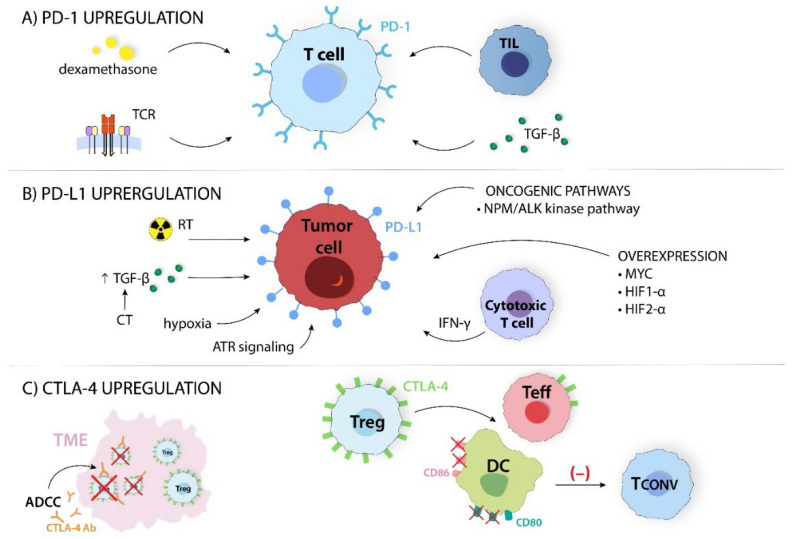
Many ICs expression mechanisms can affect the properties of immune cells. (**A**) PD-1, programmed cell death; TCR, T cell receptor; TGF-β, transforming growth factor β; TIL, tumor-infiltrating lymphocyte; (**B**) ATR, ataxia telangiectasia and RAD3-related protein; CT, chemotherapy; HIF-1α, hypoxia-inducible factor 1α; HIF-2α, hypoxia-inducible factor 2α; IFNγ, interferon γ; NPM/ALK, nucleophosmin/anaplastic lymphoma kinase; PD-L1, programmed cell death ligand; RT, radiotherapy; TGF-β, transforming growth factor β; (**C**) ADCC, antibody-dependent cellular cytotoxicity; CD80, cluster of differentiation 80; CD86, cluster of differentiation 86; CTLA-4, cytotoxic-T-lymphocyte-associated antigen 4; CTLA-4 Ab, cytotoxic-T-lymphocyte-associated antigen 4 antibody; DC, dendritic cell; Tconv, conventional T cell; Teff, effector T cell; TME, tumor microenvironment; Treg, regulatory T cell.

**Figure 6 ijms-24-14114-f006:**
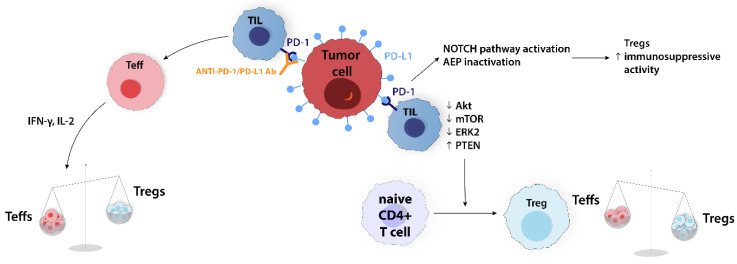
PD-1/PD-L1 axis affects Tregs and Teffs through different mechanisms. NOTCH pathway activation and AEP inactivation induce Tregs immunosuppressive activity. Downregulation of AKT, mTOR, ERK2, and upregulation of PTEN mediates the conversion of the naïve CD4+ T cells into Tregs. Moreover, blockade of PD-1/PD-L1 interaction can increase Teffs proliferation and restore their secretory function. AEP, asparaginyl endopeptidase; Akt, cellular homolog of murine thymoma virus Akt8 oncogene; anti-PD-1/PD-L1 Ab, anti-PD-1/PD-L1 antibody; ERK2, extracellular-signal-regulated kinase 2; mTOR, mammalian target of rapamycin; PD-1, programmed cell death; PD-L1, programmed cell death ligand; PTEN, phosphatase and tensin homolog deleted on chromosome 10; Teff, effector T cell; TIL, tumor-infiltrated T cell; Treg, regulatory T cell.

**Figure 7 ijms-24-14114-f007:**
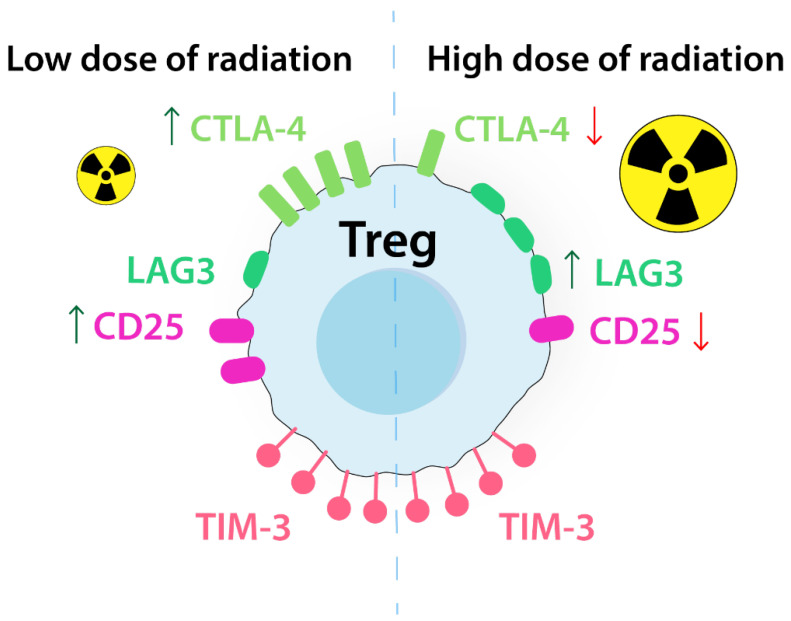
Radiation affects the expression of ICs differently depending on the dose. Low-dose radiation upregulates CTLA-4 and CD25 expression. High-dose radiation upregulates LAG-3 and downregulates CTLA-4 and CD25 expression. Radiation, regardless of the dose, upregulates TIM-3 expression. CD25, interleukin 2 receptor alpha chain; CTLA-4, cytotoxic-T-lymphocyte-associated antigen 4; LAG-3, lymphocyte activation gene-3; TIM-3, T cell immunoglobulin and mucin-domain containing-3 protein.

**Figure 8 ijms-24-14114-f008:**
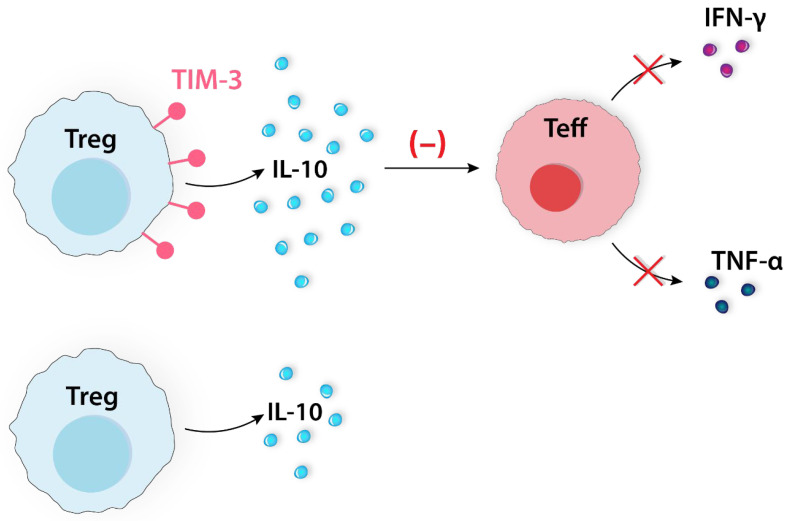
Tregs do not express TIM-3 constitutively, except the intratumoral Tregs. TIM-3-positive Tregs secrete higher amounts of IL-10 compared to TIM-3-negative Tregs. A higher level of IL-10 effectively inhibits IFN-γ and TNF-α secretion by Teffs. IFN-γ, interferon γ; IL-10, interleukin 10; Teff, effector T cell; TIM-3, T cell immunoglobulin and mucin-domain containing-3 protein; TNF-α, tumor necrosis factor α; Treg, regulatory T cell.

**Figure 9 ijms-24-14114-f009:**
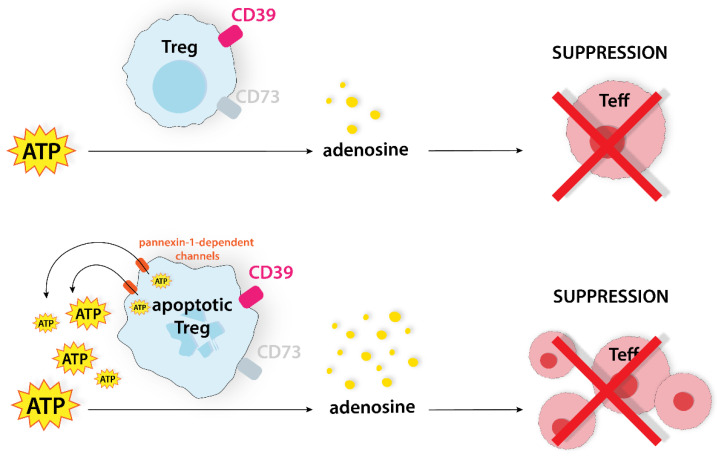
Tregs convert ATP to adenosine, suppressing Teffs. Apoptotic Tregs release more ATP through pannexin-1-dependent channels, which causes more effective suppression of Teffs. ATP, adenosine triphosphate; CD39, ectonucleoside triphosphate diphosphohydrolase-1; CD73, ecto-5′-nucleotidase; Teff, effector T cell; Treg, regulatory T cell.

**Figure 10 ijms-24-14114-f010:**
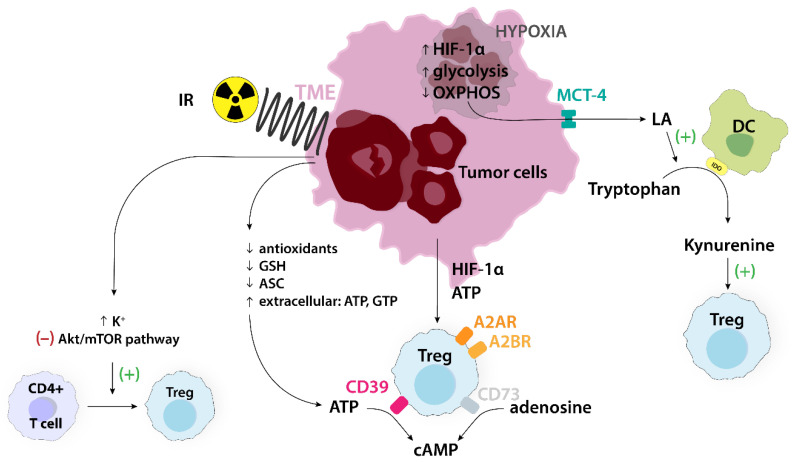
Numerous processes occur in TME that induce the immunosuppressive properties of Tregs. Hypoxia increases HIF-1α and glycolysis, and decreases OXPHOS, which enhances the amount of lactic acid (LA) in TME. LA stimulates the conversion of tryptophan into kynurenine, which stimulates Tregs. Ionizing radiation causes cancer cell degradation, which increases the amount of potassium in TME, stimulating the conversion of CD4+ T cells into Tregs. ATP, also released from tumor cells, is converted to cAMP, which leads to Tregs recruitment. A2AR, adenosine A2A receptor; A2BR, adenosine A2B receptor; Akt, cellular homolog of murine thymoma virus Akt8 oncogene; ASC, ascorbate; ATP, adenosine triphosphate; cAMP, cyclic adenosine monophosphate; CD39, ectonucleoside triphosphate diphosphohydrolase-1; CD73, ecto-5′-nucleotidase; DC, dendritic cell; GSH, glutathione; GTP, guanosine triphosphate; HIF-1α, hypoxia-inducible factor 1α; IDO, indoleamine 2,3-dioxygenase; IR, ionizing radiation; LA, lactic acid; MCT-4, monocarboxylate transporter 4; mTOR, mammalian target of rapamycin; TME, tumor microenvironment; Treg, regulatory T cell.

**Figure 11 ijms-24-14114-f011:**
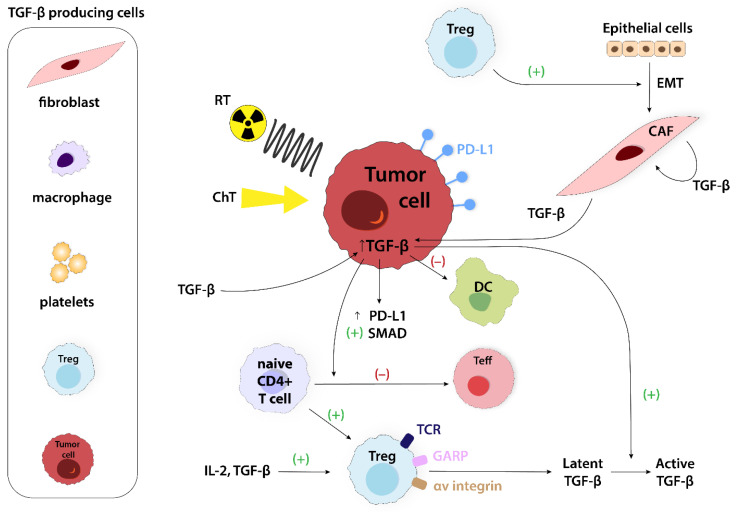
Examples of TGF-β-producing cells (left side). The role of TGF-β in TME is multimodal. TGF-β can stimulate tumor progression in advanced stages. It also promotes the conversion of naive T cells into Tregs and inhibits conversion into Teffs. TGF-β upregulates PD-L1 expression and mediates SMAD activation and abolishes antigen-presenting functions of DCs. CAF, cancer-associated fibroblast; CHT, chemotherapy; DC, dendritic cell; EMT, epithelial–mesenchymal transition; GARP, glycoprotein A repetitions predominant; IL-2, interleukin 2; PD-L1, programmed cell death ligand; RT, radiotherapy; SMAD, suppressor od mothers against decapentaplegic; TCR, T cell receptor; Teff, effector T cell; TGF-β, transforming growth factor β; Treg, regulatory T cell.

**Table 1 ijms-24-14114-t001:** Mechanisms of treatment resistance.

Reference	Type of Resistance	Factor	Mechanism
Wang et al. [23]	Primary	Reduction in IL-2 available for Teffs	IL-2 preferentially binds to CD25 on Tregs
Wang et al. [23]	Primary	Cell-bound Gal-1 ligation to GM1 on Teffs	Induces Ca2+ influx via TRPC5 channels, leading to growth arrest
Su et al. [24]	Primary	Inhibition of Teffs proliferation and IL-2 synthesis	By directly transferring inhibitory cAMP through the gap junctions of Teffs
Li et al. [25]	Primary	Induction of cytolysis of B cells, NK cells, and CD8+ T cells	In a granzyme B- and perforin-dependent manner
Li et al. [25]	Primary	Immunosuppressive cytokines such as TGF-β, IL-10, and IL-35 derived from Tregs	Inhibit the differentiation, proliferation, and functions of Teffs
Li et al. [25]	Primary	TGF-β	Promotes the conversion of activated Tconv into cells with an immunosuppressive phenotype
Li et al. [25]	Primary	LAG-3 expression on Tregs	Binds to MHC class II molecules on immature DC, blocking their maturation and limiting T-cell-mediated immune responses
Li et al. [25]	Primary	Recruitment of Src homology phosphotyrosyl phosphatase 2 (SHP2) by PD-1/PD-L1 interaction	Decreases the activity of the PI3K/Akt/mTOR signaling pathway and upregulates the expression of FoxP3, thereby inducing Tregs differentiation
Wardell et al. [26], Routy et al. [27]	Primary	CTLA-4 binding to CD80/CD86 on APCs	Limits T cell activation, inhibits T cell responses, induces a CTLA-4-mediated increase in IDO, and lowers the concentration of tryptophan, which is necessary for Teffs to proliferate
Routy et al. [27]	Primary	Kynurenine	Protects tissue from inflammation-mediated damage and participates in cancer immune escape
Tung et al. [28]	Primary	Releasing Evs, such as exosomes, stocked with miRNA	Inhibit target cells such as Teffs
Tung et al. [28]	Primary	Transferring Evs to DC with the induction of a tolerogenic phenotype	Increased immunosuppressive IL-10 and decreased IL-6 production
Schoenfeld et al. [17]	Acquired	T-cell-mediated immune activation	Antigen recognition in MHCs presented by APCs
Schoenfeld et al. [17]	Acquired	Loss of MHC class I, resulting in the loss of β2-microglobulin (B2M) and MHC class I expression and eventually an acquired defect in antigen presentation	B2M gene mutation
Schoenfeld et al. [17]	Acquired	Disorders in inducing tumor cell death	A defect of IFN-γ response triggers a signaling cascade in tumor cells through the activation of the JAK/STAT pathway that mediates MHC class I and PD-L1 expression
Schoenfeld et al. [17]	Acquired	Tumor progression despite receiving ICIs	Inactivating mutations in JAK/STAT components
Schoenfeld et al. [17]	Acquired	Increased expression of immunosuppressive cytokines and decreased IFN-γ, leading to inhibition of T-cell-mediated infiltration and immunity	Loss of the tumor suppressor PTEN
Schoenfeld et al. [17]	Acquired	The effect of WNT/β-catenin	Increased levels of immunosuppressive cytokines, promotion of Tregs, and lack of significant T cell infiltration
Schoenfeld et al. [17]	Acquired	Additional inhibitory checkpoints	Upregulated expression of various T cell checkpoints such as TIM-3, LAG-3, and VISTA

**Table 2 ijms-24-14114-t002:** Examples of mechanisms of TME metabolism involved in Treg-mediated treatment resistance.

Reference	Factor	Role in TME	Mechanism
Sebastian et al. [60], van Gisbergen et al. [61]	Aerobic glycolysis	Gain energy that supports biomass production and rapid proliferation	Altered vascularization and OXPHOS-dependent increased oxygen consumption rate cause hypoxia areas in TME with following HIF-1α upregulation
van Gisbergen et al. [61]	Activated HIF-1α	Supports the depletion of ROS and promotes angiogenesis	Promotes aerobic glycolysis and downregulates OXPHOS
Raychaudhuri et al. [62]	Lactate(glycolysis end product)	Contributes to the induction of FoxP3+ CD4+ Tregs; increases acidification of the TME	Enhances tryptophan metabolism and kynurenine production by DCs; enhances expression of MCT4
Gupta et al. [63]	Irradiation	It is associated with tumor aggressiveness and poor outcome	Induces tumor-permissive changes, such as reduced levels of antioxidants, glutathione, and ascorbate, and elevated levels of energy carriers, such as extracellular ATP and GTP
Ring et al. [64]	ATP	Strong DAMP	Activates Tregs and stimulates their suppressive function by producing immunosuppressive adenosine via the ectonucleotides CD39 and CD73
Feng et al. [57]	Hypoxia	Generates high extracellular adenosine concentration and accumulation of intracellular cAMP, which leads to the Tregs recruitment	Upregulates HIF-1α in TME cells, which then promotes CD39/CD73 and adenosine receptors A2AR and A2BR expression
Conforti [59]	Necrosis of tumor cells	Inhibits the polarization of resting CD4+ T cells into effector cells and promotes the Tregs development	It results in an increased extracellular potassium concentration, which subsequently suppresses the Akt/mTOR pathway

**Table 3 ijms-24-14114-t003:** Mechanisms of TGF-β-dependent upregulation of Tregs.

Reference	Factor	Role	Mechanism
Oshimori et al. [71]	TGF-β during tumorigenesis	TGF-β-responding tumor cells are responsible for drug resistance and tumor recurrence	Induces apoptosis and inhibits the proliferation of cancer cells; in advanced stages, it can stimulate tumor progression
Neel et al. [72,73], Lainé et al. [72,73]	Tregs and TGF-β mutual relation	Promotes tumor immune escape	(1) Tregs development requires at least TCR stimulation and the TGF-β and IL-2 supply; (2) Tregs secrete latent form of TGF-β; (3) Treg cell-integrin expression is essential to activate TGF-β produced by cancer cells.
Funaki et al. [34]	Chemotherapy	CD8+ T lymphocytes repression	Increases TGF-β levels with following enhanced PD-L1 expression on cancer cells; elevates EMT markers
Funaki et al. [34], Quan et al. [74]	EMT	Contributes to stem cell generation, anticancer drug resistance, genomic instability, and localized immunosuppression	Elevated levels of E-cadherin and vimentin
Wang et al. [75]	TGF-β and the JAK/STAT axis	Promotes radioresistance	Induces the EMT, cancer stem cells, and cancer-associated fibroblasts
Liu et al. [76]	ROS	Metastasis	Mediate TGF-induced EMT

## Data Availability

Not applicable.

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
