# Peer review of "The Role of Regulatory T Cells in Cancer Treatment Resistance"

_ijms, 2023, doi:10.3390/ijms241814114_

Round 1

Reviewer 1 Report

The authors extensively reviewed the crucial role of Tregs in antitumor response. I have a few addressable concerns.

Minor concerns: 

1)    Abstract “Tregs-mediated” to “Treg-mediated”

2)    Introduction “…through kynurenine production and immune checkpoints (ICs) expression” to “…through kynurenine production and immune checkpoint (IC) protein expression”. Immune checkpoints are not a molecule unto themselves so it needs to be specified what molecule.

3)    Line 51: Extracellular vesicles (EVs) not extracellular vehicles. 

4)    Gene names should be in the same format throughout the manuscript.

Major Concerns:

1)    The authors need to be more clear about which cancer when describing studies, immunotherapies, and resistance. Most of these drugs are not used in every type of cancer and furthermore, they may have very different effectiveness in different cancers. Also, acquired resistance may occur much more quickly in one cancer than another in relation to the same drug.

2)    Line 94: Authors mentioned a correlation between the accumulation of FOXP3+ Tregs and poor prognosis in various cancer. Was the correlation tested statistically? Is this a statistically significant correlation? Did they observe any difference in correlation between cancers? 

There are a few grammatical errors but this manuscript does not require a rewrite or extensive language editing.

Author Response

Thank you for your time spent on manuscript evaluation and comments. 

Minor concerns: 

1)    Abstract “Tregs-mediated” to “Treg-mediated” 

2)    Introduction “…through kynurenine production and immune checkpoints (ICs) expression” to “…through kynurenine production and immune checkpoint (IC) protein expression”. Immune checkpoints are not a molecule unto themselves so it needs to be specified what molecule. 

3)    Line 51: Extracellular vesicles (EVs) not extracellular vehicles.

4)    Gene names should be in the same format throughout the manuscript.

All minor concerns were addressed in the manuscript.

Major Concerns:

1)    The authors need to be more clear about which cancer when describing studies, immunotherapies, and resistance. Most of these drugs are not used in every type of cancer and furthermore, they may have very different effectiveness in different cancers. Also, acquired resistance may occur much more quickly in one cancer than another in relation to the same drug.

Detailed information about the type of cancer were added when possible.

2)    Line 94: Authors mentioned a correlation between the accumulation of FOXP3+ Tregs and poor prognosis in various cancer. Was the correlation tested statistically? Is this a statistically significant correlation? Did they observe any difference in correlation between cancers? 

Detailed information on these results was added based on the primary publication.

Reviewer 2 Report

Anna et al. delivers a comprehensive overview of Treg in cancer resistance. Th review is meticulously structured, highly accessible and skillfully composed, greatly aiding researchers in swiftly grasping the subject matter. However, the figures are not found in the submission. 

Minor revision: Anna and her colleagues present a comprehensive review on Treg in cancer treatment resistance, with a particular focus on immune checkpoint-associated resistance. The manuscript demonstrates a well-structured approach, enhanced by meticulous referencing and clear diagrams that expound upon the molecular mechanism and functional roles of Tregs in the context of cancer therapy. The paper imparts perceptive insights that facilitate a swift understanding of the pivotal advancements within Treg studies. One suggestion for enhancement is to integrate a succinct discussion segment that delves into the existing correlation between Tregs and macrophages, as both are recognized as crucial architects of tumor-related immunosuppression.

Author Response

Thank you for your time spent evaluating our manuscript.

Minor revision: Anna and her colleagues present a comprehensive review on Treg in cancer treatment resistance, with a particular focus on immune checkpoint-associated resistance. The manuscript demonstrates a well-structured approach, enhanced by meticulous referencing and clear diagrams that expound upon the molecular mechanism and functional roles of Tregs in the context of cancer therapy. The paper imparts perceptive insights that facilitate a swift understanding of the pivotal advancements within Treg studies. One suggestion for enhancement is to integrate a succinct discussion segment that delves into the existing correlation between Tregs and macrophages, as both are recognized as crucial architects of tumor-related immunosuppression.

A paragraph on existing correlation between Tregs and macrophages was added to the manuscript.